# Taxonomic Review of the Genus *Caloptilia* Hübner, 1825 (Lepidoptera: Gracillariidae) with Descriptions of Three New Species and Seven Newly Recorded Species from Korea

**DOI:** 10.3390/insects13121107

**Published:** 2022-11-30

**Authors:** Da-Som Kim, Young-Min Shin, Ji-Young Lee, Bong-Kyu Byun

**Affiliations:** 1Basic Science Division, National Science Museum of Korea, Daejeon 34143, Republic of Korea; 2Division of Forest Biodiversity, Korea National Arboretum, Pocheon 11816, Republic of Korea; 3Department of Biological Science and Biotechnology, Hannam University, Daejeon 34054, Republic of Korea

**Keywords:** Lepidoptera, Gracillariinae, new species, newly recorded species, leafminer

## Abstract

**Simple Summary:**

Caloptilia is a genus of the family Gracillariidae, one of the most diverse groups of Microlepidoptera, with more than 2000 described species worldwide. The species of this family are known as leafminers because they mine into the leaves of various host plants. *Caloptilia* has been poorly studied in Korea, with 50% fewer known species than those in the neighboring countries, viz., China, Russia, and Japan. In this study, we aimed to clarify fauna of the genus *Caloptilia* in Korea by describing three new species and seven newly recorded species in the country. In addition, we list all known species from Korea, along with images of the adult specimens and genitalia.

**Abstract:**

In this study, 29 species of *Caloptilia* Hübner, 1825, belonging to the family Gracillariidae, were recognized in Korea. Among these, three species, i.e., *C. purpureus* sp. nov., *C. koreana* sp. nov., and *C. xanthos* sp. nov., are described as new to science. In addition, seven species of this genus are reported for the first time in Korea. All known species were enumerated, based on their available information. Adult specimens and genitalia of the new and newly recorded species were examined and described using all available information.

## 1. Introduction

The family Gracillariidae is one of the most diverse groups of Microlepidoptera, with more than 2000 described species belonging to 112 genera worldwide [1]. The genus *Caloptilia* is the second largest group in the family, comprising more than 450 species worldwide [1]. They are cosmopolitan in distribution, with 180 species, ca. 40% of the total known species, found in the Palearctic region. They can be distinguished from the other genera based on the following observations (i) adults in a resting posture raising their head upward; (ii) abdomens (A) with a membranous tergite and sclerotized sternite on A8; (iii) females with a pair of signa; (iv) final instar larvae with leaf rolling behavior that pupate within the leaves of host plants [2,3]. 

This study is part of a larger study on this family that was recently conducted in Korea to update the faunistic information for 91 species of 25 genera under seven subfamilies [4,5,6,7,8,9,10,11,12,13,14,15,16,17,18].

Among them, the genus *Caloptilia* was not well studied until Shin et al. [5] enumerated 19 species from Korea, with four newly reported species. Of them, two species, *C. fidella* and *C. illicii*, were misidentified as Gracillaria albicapitata and C. sapporella, respectively. Recently, Lee and Jeun (2022) [16] reported two newly recorded species: *C. pyrrhaspis* (Meyrick) and *C. syrphetias* (Meyrick). To date, 21 species of this genus have been reported in this country. 

In this study, a more intense investigation of the genus *Caloptilia* was conducted, using the results of an examination of the big collection from the 1970s. We found three new species, *C. purpureus* sp. nov., *C. koreana* sp. nov., and *C. xanthos* sp. nov., and recorded seven species for the first time in Korea. These include *C. acericola* Kumata, 1966 [19], *C. celtidis* Kumata, 1982 [2], *C. dentata* Liu and Yuan, 1990 [20], *C. kadsurae* Kumata, 1966 [18], *C. monticola* Kumata, 1966 [18], *C. recitata* (Meyrick, 1918) [21], and *C. soyella* (van Deventer, 1904) [22]. Here, we provide a description of these three new species along with the taxonomic arrangements and annotated checklist of all known species. In addition, information on collection locality, host plant, and distributional range is provided. 

## 2. Materials and Methods

The specimens examined in this study were deposited at the Systematic Entomology Laboratory of Hannam University, Daejeon, Korea (HNSUEL). Male and female genitalia were dissected and mounted with Euparal solution, following the procedure described by Holloway et al. [23]. Images of the adults were taken using a digital camera (Canon EOS 600D, Canon Inc., Ota, Tokyo, Japan), and images of the genitalia were captured using a digital camera attached to a microscope, LEICA M205C (Leica Microsystems, Wetzlar, Hesse, Germany) and refined using Photoshop CS5 software. Terminology for the genitalia of both sexes was followed after Landry [24].

## 3. Results

### 3.1. Description of New Species

#### 3.1.1. *Caloptilia purpureus* sp. nov. Kim and Byun, 2022 Bora-ga-neun-na-bang

LSID. urn:lsid:zoobank.org:act:7985CE24-A706-4AF9-B8E5-55405021EDAC

Type. Holotype. [KOREA] [JN] 1♂, Mt. Geumo, Dolsan-eup, Yeosu-si, 10 October 2018 (emergence data: 28 October 2018) (leg. UH Heo), gen. slide no. HNUSEL-5592-coll. UH Heo; Paratype. [JN] 1♀, Mt. Geumo, Dolsan-eup, Yeosu-si, 10 October 2018 (emergence data: 1 November 2018) (leg. UH Heo), gen. slide no. HNUSEL-5591-coll. UH Heo.

Diagnosis. This species is similar to *C. theivora* in appearance, but can be distinguished by the costal margin with narrow yellowish streak after the blotch apically and metallic purple ground color of forewing. In the genitalic structures, the new species is different from the latter by having numerous long setae on apex of valva in male genitalia and two signa on corpus bursae in female genitalia.

Adult (Figure 1a). Forewing length 3.5 mm. Head grayish ochreous and tinged with violet reflection; frons pale yellowish white and face yellow; maxillary palpus yellow with a tiny fuscous spot on the median part; labial palpus yellow with apex fuscous; antenna pale ochreous and scape fuscous. Thorax pale grayish fuscous; fore legs fuscous with whitish tarsus; hind femur yellow with a fuscous median broad band; hind tibia and tarsus pale ochreous. 

Forewing ground color grayish fuscous with violet reflection; a triangular goldish yellow blotch, covering nearly half of forewing, extends to wing fold with blackish edges and 2–3 blackish dots on apical margin; a goldish yellow streak begins just after blotch apically, occupying 1/4 of forewing and broader at median without any edges; veins with Sc and R1 close, R3 and R4 very close, M2 stalked with M3. Hindwing is similar to *C. koreana*.

Male genitalia (Figure 2a). Tegumen ovate with parallel side to apex and as long as vinculum or slightly longer than vinculum. Valva up-curved, elongated, rectangular shape with parallel side from base to 1/3 of valva, dilated on apex, dorsal margin more elongated on apex and apical margin straight; long and slender setae along apical margin to a half of ventral margin, more sparse ventrally, more short and strong setae sparse along the apical margin. Vinculum triangular shape with acute apex. Juxta small and ‘u’ shaped. Aedeagus bar-shaped, apex somewhat rectangular shape, slightly sclerotized and slightly bent inward without any cornuti.

Female genitalia (Figure 3a). *Papillae anales* moderated, long setae on vertex sparser with a pair of long scales on each, laterally of base; posterior apophyses triangular shaped broadening to base and as long as anterior. Ostium bursae small in opening size; sternum VIII divided into two lateral parts and shot as long as 1/6 of posterior apophyses. Dustus bursae very slender, as long as 2.5 times of corpus bursae and coiled near corpus bursae. Corpus bursae ovate with two signa on median; signa with different size each other, blunt and rather rectangular apex. 

Etymology. The specific name is derived from the color of forewing, purple. 

Distribution. Korea (endemic, Yeosu-si). 

Host plants. *Sageretia theezans* (L.) Brongn. [Rhamnaceae] in Korea (in this study). 

Remarks. This species was reared from *Sageretia theezans* (L.) Brongn. of the family Rhamnaceae in this study.

The triangular yellow costal blotch of the new species almost the same as *C. theivora*; however, costal margin with narrow yellowish streak after the blotch apically. In the new species, long setae on apical of valva dense along margin, whereas *C. theivora* naked at median of apical margin. In female genitalia, there are two signa on corpus bursae in new species, while *C. theivora* has only a signum. 

#### 3.1.2. *Caloptilia koreana* sp. nov. Kim and Byun, 2022 Han-guk-ga-neun-na-bang 

LSID. urn:lsid:zoobank.org:act:EFDBC3D3-71BA-4ED2-9391-28991F5E14A1

Type. Holotype. [KOREA] [GG] 1♀, Lake Yuklim, Gwangneung, 7 September 2017 (leg. Lim, Kim, Lee, Shin, Roh), gen. slide no. HNUSEL-5575-coll. KNAE.

Diagnosis. This species is similar to *C. chrysolampra*, but can be distinguished by the more extended size of the yellow costal blotch of the forewing and female genitalia with well sclerotized ductus bursae in the new species, whereas they are very slender and membranous in the latter. 

Adult (Figure 1b). Forewing length 3 mm in female. Head grayish ochreous and occiput gray with flat and large scales; frons white and face pale yellowish white; maxillary palpus white with a tiny fuscous spot on the first segment laterally; labial palpus silvery white with a tiny fuscous spot laterally and fuscous apically; each segment of antenna grayish fuscous, pale ochreous basally. Thorax grayish with a pale ochreous median line; tegular more darkened to fuscous posteriorly; fore and middle legs fuscous with whitish tarsus; hind legs pale ochreous with tiny fuscous spots apically.

Forewing ground color grayish fuscous, tinged with violet reflection; a yellow dorso-basal streak from base to 1/6 of forewing; large and rounded yellow blotch begins at 1/3 to base and extends to wing fold, the interval between blotch and dorsal margin broadens more apically and tiny blackish dots sparsely on costal margin; venation with R_1_ arising from before middle, R_2_ arising from 2/3 of cell, R_3_, R_4_ and R_5_ not close, M2 and M3 stalked, M3 and CuA close near discal end of cell. Hindwing lanceolate and ground color silvery dark gray; vein with Rs reaching to subapex of costa, M1 and M2 form a branch, M3 distant from M2, then close to Cu.

Male genitalia. Unknown.

Female genitalia (Figure 3b). Papillae anales somewhat acute on vertex, short and sclerotized on base; posterior apophyses broaden at base and slightly sclerotized; anterior apophyses as long as posteriores. Lamella postvaginalis with a well sclerotized flap, the flap reversed pot shaped; ostium bursae with same width as ductus bursae.

Ductus bursae strongly sclerotized with many horizontal wrinkles, as long as corpus bursae, slightly curved to small ‘S’ shape and broadened to near the corpus bursae. Corpus bursae ellipse form and slightly sclerotized near ductus bursae with two signa on 1/4 of corpus bausae; signa falciform, somewhat straight, and acute apically. 

Etymology. The specific name is derived from the type locality, Korea.

Distribution. Korea (endemic, Gwangneung).

Host plants. Unknown.

Remarks. This species is similar to *C. chrysolampra*, but can be distinguished by the size of yellow costal blotch of the forewing. The yellow costal blotch of new species extends more to nearly apex, and forewing ground color grayish fuscous. In female genitalia, ductus bursae well sclerotized with a pair of falciform shaped two signa. 

#### 3.1.3. *Caloptilia xanthos* sp. nov. Kim and Byun, 2022 No-rang-jeom-ga-neun-na-bang

LSID. urn:lsid:zoobank.org:act:F542A98A-295C-4BB0-BC8B-96CD6BB15CE4

Type. Holotype. [KOREA] [JN] 1♀, Namseong-ri, Cheonggye-myeon, Muan-gun, 3 July 2019 (leg. BK Byun), gen. slide no. HNUSEL-5602-coll. HNUSEL.

Diagnosis. This species is similar to *C. theivora*, but can be distinguished by rather long corpus bursae with two signa in female genitalia

Adult (Figure 1c). Forewing length 4 mm. Head covered with pale grayish fuscous. Thorax grayish fuscous with two ochreous lateral line on tegular; legs fuscous and white; middle tarsus white with a fuscous median band. Forewing purplish fuscous with a yellow costal blotch; a yellow costal blotch begins at 1/3 to base and is triangular with very acute apex without edges; venation with R_1_ arising from 1/3 of cell, R_3_, R_4_ and R_5_ not close, M1 distant from M2+M3 at base, M2 and M3 stalked, M3 and CuA not close. Hindwing lanceolate and ground color silvery dark gray; veins with Rs reaching to apex of costa, M1 and M2 form a branch, M3 distant from M2.

Male genitalia. Unknown.

Female genitalia (Figure 3c). Papillae anales slightly swollen with rounded process on vertex; posterior apophyses as long as anterior and slightly broadened at base. Ostium bursae on a sclerotized ellipse form disc and small in opening size; antrum slightly sclerotized and reduced. Ductus bursae as long as 1.7 times of corpus bursae and slender. Corpus bursae elongated, very long and slender with a lot of minute spinules on whole surface; signa dislocated from each other, well sclerotized and base more elongated to inner side. 

Etymology. The specific name is derived from the yellowish patch of forewing. 

Distribution. Korea (endemic, Muan-gun). 

Host plants. Unknown. 

Remarks. This species is similar to *C. theivora*, but the new species has two signa on corpus bursae of female genitalia, whereas the latter has a signum. This species is similar to the members of the genus *Caloptilia* in appearance. In the new species, the yellow triangular costal blotch in the forewing is almost the same as *C. theivora*; however, it can be distinguished by the small yellow spots along the costal margin from the end of blotch to near the subapex, then tinged with yellowish scales toward apex. In female genitalia, there are two singa on corpus bursae, instead of only one sigum in the latter.

### 3.2. Redescription of New Records

#### 3.2.1. *Caloptilia acericola* Kumata, 1966 Gin-no-lang-ga-neun-na-bang

*Caloptilia acericola* Kumata, 1966: 2–3 [19]. TL: Hokkaido, Japan. TD: EIHU (Holotype; Paratypes). 

Adult (Figure 4a). Forewing length 5 mm. 

Male genitalia (Figure 2b). 

Female genitalia (Figure 5a). 

Material examined. [GG] 1♀, Korea National Arboretum, Gwangneung, 20 May 2008 (leg. SY Park, BW Lee, SR Kim, DH Kwon), gen. slide no. HNUSEL-5577-coll. KNAE; [GW] 3♂, 2♀, Mt. Odae, 27 May 1991 (leg. BK Byun), gen. slide no. NAK-533, HNUSEL-5603, 5604-coll. HNUSEL; 1♂, Mt. Odaesan, Jinbu-Myeon, Pyeongchang, 20 June 2012 (leg. SY Park, JO Lim, JS Lim), gen. slide no. HNUSEL-5608-coll. KNAE; [CB] 1♂, Mt. Minjuji, Maegok-myeon, Yeongdong-gun, 22 May 2014 (leg. IJ Choi, JW Nam, MH Kim), gen. slide no. HNUSEL-5571-coll. KNAE; 1♂, Namwon-si, Sannae-myeon, Jeongnyeongchi 23 May 2012 (leg. NH Ahn, JJ Park), gen. slide no. HNUSEL 5581-coll. NIBR; [GN] 1♂, Namhae, 1 June 1994 (leg. BK Byun), gen. slide no. HNUSEL-5552-coll. HNUSEL. 

Distribution. Korea (new record), Japan, Russia. 

Host plants. *Acer japonicum* Thunb., *A. palmatum* Thunb., *A. pictum* Thunb. ex Murray [Sapindaceae] in Japan [1,19]. *A. pictum* Thunb. ex Murray, *A. pseudosieboldianum* (Pax) Kom. [Sapindaceae] in Russia [1,25,26]. 

Remarks. This species has been confused with *C. aceris*; however, it can be distinguished with the presence of a minute spinule on ductus bursae of female genitalia, which is absent in the latter. 

#### 3.2.2. *Caloptilia celtidis* Kumata, 1982 Huin-mu-nui-ga-neun-na-bang 

*Caloptilia celtidis* Kumata, 1982: 76–79 [2]. TL: Honshu, Japan. TD: EIHU (Holotype); EIHU, BMNH (Paratypes). 

Adult (Figure 4b). Forewing length 5 mm. Two forms with aestival and autumnal. 

Male genitalia (Figure 2c). 

Female genitalia (Figure 5b).

Material examined. [JN] 2♀, Mt. Wangui, Haeryong-myeon, Suncheon-si. 30 July 2016 (emergence date: 10 August 2016), gen. slide no. HNUSEL-5613-coll. UH Heo; 1♀, experiment forest of Seoul University, Gwangyang-si, 13 September 2016 (emergence date: 29 September 2016)-coll. UH Heo; 1♂, Mt. Jogye, Suncheon-si, 4 July 2018 (leg. BK Byun), gen. slide no. HNUSEL-5561-coll. HNUSEL. 

Distribution. Korea (new record), China, Hong Kong, Japan. 

Host plants. *Celtis sinensis* Persoon [Cannabaceae] in Korea (in this study). *C. sinensis* Persoon [Cannabaceae] in China [1,20]. *C. jessoensis* Koidz., *C. sinensis* Persoon [Cannabaceae] in Japan [1,2].

Remarks. This species was reared from *Celtis sinensis* Persoon of the family Cannabaceaee in this study. 

#### 3.2.3. *Caloptilia dentata* Liu & Yuan, 1990 Ne-mo-mu-nui-ga-neun-na-bang

*Caloptilia (Caloptilia) dentata* Liu & Yuan, 1990: 186 [20]. TL: Beijing, China. TD: IZAS (Holotype; Paratypes). 

Adult (Figure 4c). Forewing length 5 mm. 

Male genitalia (Figure 2d). 

Female genitalia. see Liu & Yuan [20]. 

Material examined. [JB] 2♂, 1♀, Jangan-ri, Songgwang-myeon, Suncheon-si, 4 July 2018 (leg. BK Byun), gen. slide no. HNUSEL-5504-coll. HNUSEL; 2♀, Seji-myeon, Naju-si, 31 July 2018 (leg. BK Byun)-coll. HNUSEL; [GN] 1♂, Mt. Keum, Namhae, 7 August 1982 (leg. KT Park), gen. slide no. HNUSEL-5473-coll. HNU

Distribution. Korea (new record), China. 

Host plants. *Acer truncatum* Bunge [Sapindaceae] in China [1,20].

#### 3.2.4. *Caloptilia kadsurae* Kumata, 1966 Nam-o-mi-ja-ga-neun-na-bang

*Caloptilia kadsurae* Kumata, 1966: 19 [19]. TL: Honshu, Japan. TD: EIHU (Holotype; Paratypes).

Adult (Figure 4d). Forewing length 5.2 mm.

Male genitalia (Figure 2e). 

Female genitalia (Figure 5c). 

Material examined. [JN] 2♂, Wando Arboretum, Wando, 31 August 2019 (emergence date: 16 September 2019) (leg. UH Heo), gen. slide no. HNUSEL-5593, 5594-coll. UH Heo; [JJ] 1♀, Sanghyo-dong, Seogwipo-si, 24 July 2019 (leg. BK Byun), gen. slide no. HNUSEL-5558-coll. HNUSEL. 

Distribution. Korea (new record), Japan. 

Host plants. *Kadsura japonica* Dunal [Magnoliaceae] in Korea (in this study). *K. japonica* Dunal [Magnoliaceae] in Japan [1,19].

Remarks. This species was reared from *K. japonica* Dunal of the family Magnoliaceae in this study.

#### 3.2.5. *Caloptilia monticola* Kumata, 1966 Oe-ban-jeom-ga-neun-na-bang

*Caloptilia monticola* Kumata, 1966: 8 [19]. TL: Honshu, Japan. TD: EIHU (Holotype; Paratypes). 

Adult (Figure 4e). Forewing length 6 mm.

Male genitalia. see Kumata 1966 [19].

Female genitalia (Figure 5d). 

Material examined. [GB] 1♀, Mt. Tonggo, Geumgangsong-myeon, Uljin-gun, 23 June 2017 (leg. BS Park, SM Na, DJ Lee), gen. slide no. HNUSEL-5587-coll. INU. 

Distribution. Korea (new record), China, Japan, Russia. 

Host plants. *Acer argutum* Maxim., *A. ginnala* Maxim., *A. pentaphyllum* Diels., *A. pictum* Thunb. ex Murray, *A. rufinerve* Siebold & Zucc., *A. semenovii* Regel & Herd, *A. ukurunduense* Trautv. & Mey. [Sapindaceae] in China [1,20,27]. *A. argutum* Maxim., *A. distylum* Siebold & Zucc., *A. ginnala* Maxim., *A. micranthum* Siebold & Zucc., *A. pictum* Thunb. ex Murray, *A. rufinerve* Siebold & Zucc., *A. tschonoskii* Maxim., *A. ukurunduense* Trautv. & Mey. [Sapindaceae] in Japan [1,2,19]. *A. pictum* Thunb. ex Murray, *A. semenovii* Regel & Herd, A. sp., *A. ukurunduense* Trautv. & Mey. [Sapindaceae] in Russia [1,28,29].

#### 3.2.6. *Caloptilia recitata* (Meyrick, 1918) Keun-gal-saeg-mu-nui-ga-neun-na-bang 

*Gracillaria recitata* Meyrick, 1918b: 178–179 [21]. TL: Assam, India. TD: BMNH (Syntypes). 

*Caloptilia recitata*: Issiki, 1957: 30 [30]. 

Adult (Figure 4f). Two forms with aestival and autumnal.

Aestival: Forewing orange brown with yellow blotch; basal part grayish brown; a yellow costal blotch extends along costa margin near apex, a half of blotch to base broadens to wing fold, back to narrow and a row of tiny black spots along costal margin within blotch; a black spot on the center of forewing near the boundary line of the yellow blotch; cilia brown on dorsal margin and blackish gray on tornus. 

Autumnal: Forewing ground color purplish brown; basal-costa margin dark brown; a boundary line of yellow blotch more obscure than aestival form; a row of tiny blackish spots along costa margin, a spot on 2/3 to base, which is more distinct and larger. 

Male genitalia (Figure 2f). Tegumen concave to inner surface, as long as 2/3 of vinculum and apex round. Valva moderate in shape with other species of this group; spatulate shaped, base narrow and broadens to apex with slender scales densely. Vinculum elongated, triangular and apical round. Aedeagus slender, bar-shaped and narrowed to apex; a long and slender cornuti on apical part, three of short and stout cornuti near base and the last one more elongated with bifurcated apical part.

Female genitalia (Figure 5e). Well sclerotized except for corpus bursae. posterior apophyses as long as anteriores. Ostium bursae moderate opening size, antrum with two sclerotized projections on both sides. Ductus bursae 2 times longer than corpus bursae, tubular, narrows caudally, broadens again to corpus bursae and slightly twists near corpus bursae. Corpus bursae membranous except for connection part with ductus bursae, elongated and slightly hollowed medially; two signa above median part and rapidly curved inside. 

Material examined. [DJ] 1♂, Daedeok, Jeonmin-dong, 5 June 2015 (leg. BK Byun), gen. slide no. HNUSEL-5392-coll. HNUSEL; [JN] 1♂, Chusan-ri, Ongryong-myeon, Gwangyang, 2 June 2018 (leg. BK Byun), gen. slide no. HNUSEL-5560-coll. HNUSEL; [JB] 4♂, 4♀, Sangseo-myeon, Buan-gun, 29 September 2018 (leg. BK Byun), gen. slide no. HNUSEL-5543, 5544-coll. HNUSEL. 

Distribution. Korea (new record), China, Hong Kong, Japan, India, Nepal. 

Host plants. *Cotinus coggygria* Scop., *Rhus javanica* L., *Toxicodendron sylvestre* (Siebold & Zucc.) Kuntze, *T. trichocarpum* (Miq.) Kuntze in China [Anacardiaceae] [1,20]. *R. javanica* L., *T. sylvestre* (Siebold & Zucc.) Kuntze, *T. trichocarpum* (Miq.) Kuntze in Japan [Anacardiaceae] [1,2]. *R. javanica* L. in Nepal [Anacardiaceae] [1,2].

Remarks. This species was collected from *Ailanthus altissima* (Mill.) [Simaroubaceae] with pupal cocoon on the back side. 

#### 3.2.7. *Caloptilia soyella* (van Deventer, 1904) Tob-ni-ga-neun-na-bang

*Gracillaria soyella* van Deventer, 1904: 22–25 [22]. TL: Java, Indonesia. TD: RNHL (Lectotype; Paralectotype). 

*Caloptilia soyella*: Issiki, 1950: 451 [31]. 

Adult (Figure 4g). Forewing length 4.2 mm. 

Male genitalia (Figure 2g). 

Female genitalia (Figure 5f,g). Papillae anales moderate with slender scales on apex; posterior apophyses 1.5 times longer than anteriores; sterigma well sclerotized; ostium bursae with ‘M’ shaped sclerotized lobe on center. Ductus bursae membranous, slender and 4 times longer than apophyses posteriores. Corpus bursae ovate with two sclerotized signa; each signum placed symmetrically, curved inner side and more narrowed apically. 

Material examined. [JN] 1♀, Mt. Jeseok, Boseong-gun, 15 September 2016 (emergence date: 29 September 2016), (leg. UH Heo), gen. slide no. HNUSEL-5597-coll. UH Heo; 2♂, Seji-myeon, Naju-si, 31 July 2018 (leg. BK Byun), gen. slide no. HNUSEL-5556, 5557-coll. HNUSEL. 

Distribution. Korea (new record), China, Japan, India, Indonesia, Viet Nam, Fiji, Cape Verde. 

Host plants. *Lespedeza cyrtobotrya* Miq. [Fabaceae] in Korea (in this study). *Cajanus cajan* (L.) Millsp., *Glycine max* (L.) Merr., *Kummerovia striana* Schindler, *Lespedeza cyrtobotrya* Miq., *Phaseolus mungo* L., *Vigna angularis* (Willd.) Ohwi & H. Ohashi in China [Fabaceae] [1,20]. *G. max* (L.) Merr., *K. striana* Schindler, *L. cyrtobotrya* Miq., *P. calcaratus* Roxb., *V. angularis* (Willd.) Ohwi & H. Ohashi in Japan [Fabaceae] [1,2]. *C. cajan* (L.) Millsp., *P. mungo* L. in India [Fabaceae] [1,32]. *Soya hispida* Moench in Indonesia [Fabaceae] [1,22]. 

Remarks. This species was reared from *Lespedeza cyrtobotrya* Miq. of the family Fabaceae in this study. 

### 3.3. Checklist of the Genus Caloptilia in Korea 

Genus *Caloptilia* Hübner, 1825 

*Caloptilia* Hübner, 1825: 427 [33]. 

Type species: *Tinea upupaepennella* Hübner, 1796 [34]. 

*Antiolopha* Meyrick, 1894 [35].

*Calliptilia* Agassiz, 1847 [36].

*Cecidoptilia* Kumata, 1982 [2].

*Coriscium* Zeller, 1839 [37].

*Minyoptilia* Kumata, 1982 [2].

*Ornix* Treitschke, 1833 [38].

*Phylloptilia* Kumata, 1982 [2].

*Poeciloptilia* Hübner, 1825 [34].

*Rhadinoptilia* Kumata, 1982 [2].

*Sphyrophora* Vári, 1961 [39].

*Timodora* Meyrick, 1886 [40].

#### 3.3.1. *Caloptilia acericola* Kumata, 1966 Gin-no-lang-ga-neun-na-bang

*Caloptilia acericola* Kumata, 1966: 2–3 [19]. TL: Hokkaido, Japan. TD: EIHU (Holotype; Paratypes). 

Forewing length 5 mm. 

Material examined. [GG] 1♀, Korea National Arboretum, Gwangneung, 20 May 2008 (leg. SY Park, BW Lee, SR Kim, DH Kwon), gen. slide no. HNUSEL-5577-coll. KNAE; [GW] 3♂, 2♀, Mt. Odae, 27 May 1991 (leg. BK Byun), gen. slide no. NAK-533, HNUSEL-5603, 5604-coll. HNUSEL; 1♂, Mt. Odaesan, Jinbu-Myeon, Pyeongchang, 20 June 2012 (leg. SY Park, JO Lim, JS Lim), gen. slide no. HNUSEL-5608-coll. KNAE; [CB] 1♂, Mt. Minjuji, Maegok-myeon, Yeongdong-gun, 22 May 2014 (leg. IJ Choi, JW Nam, MH Kim), gen. slide no. HNUSEL-5571-coll. KNAE; 1♂, Namwon-si, Sannae-myeon, Jeongnyeongchi 23 May 2012 (leg. NH Ahn, JJ Park), gen. slide no. HNUSEL 5581-coll. NIBR; [GN] 1♂, Namhae, 1 June 1994 (leg. BK Byun), gen. slide no. HNUSEL-5552-coll. HNUSEL. 

Distribution. Korea (new record), Japan, Russia. 

Host plants. *Acer japonicum* Thunb., *A. palmatum* Thunb., *A. pictum* Thunb. ex Murray [Sapindaceae] in Japan [19]. *A. pictum* Thunb. ex Murray, *A. pseudosieboldianum* (Pax) Kom. [Sapindaceae] in Russia [1,25,26]. 

#### 3.3.2. *Caloptilia aceris* Kumata, 1966 Dan-pung-ip-ga-neun-na-bang

*Caloptilia aceris* Kumata, 1966: 1 [19]. TL: Hokkaido, Japan. TD: EIHU (Holotype; Paratypes). 

Forewing length 5.2 mm. 

Material examined. [DJ] 1♂, Mt. Manin, Haso-dong, Dong-gu, 26 May 2018 (leg. BK Byun), gen. slide no. HNUSEL-5579-coll. HNUSEL. 

Distribution. Korea, China, Japan, Russia. 

Host plants. *Acer miyabei* Maxim., *A. palmatum* Thunb., *A. pictum* Thunb. ex Murray, *A. saccharum* Marschall [Sapindaceae] in China [1,27]. *A. miyabei* Maxim., *A. palmatum* Thunb., *A. pictum* Thunb. ex Murray, *A. saccharum* Marschall [Sapindaceae] in Japan [1,2,19]. *A. pictum* Thunb. ex Murray [Sapindaceae] in Russia [1,29].

#### 3.3.3. *Caloptilia alni* Kumata, 1966 O-li-na-mu-ga-neun-na-bang

*Caloptilia alni* Kumata, 1966: 12 [19]. TL: Hokkaido, Japan. TD: EIHU (Holotype; Paratypes). 

Forewing length 7 mm. 

Material examined. [GW] 1♂, Chuncheon-Dam, 24 July 1991 (leg. KT Park & BK Byun), gen. slide no. HNUSEL-5044-coll. HNUSEL; 1♀, Mt. Chiak, Wonju-city, 24 April 1998 (leg. Bae, Paek, Lee, Ahn), gen. slide no. HNUSEL-5048-coll. INU. 

Distribution. Korea, China, Japan, Russia. 

Host plants. Alnus hirsuta Turcz., *A. japonica* (Thunb.) Steud. in China [Betulaceae] [1,20]. *A. hirsuta* Turcz., *A. japonica* (Thunb.) Steud. in Japan [Betulaceae] [1,19]. *A. hirsuta* Turcz., *A. japonica* (Thunb.) Steud. in Russia [Betulaceae] [1,28]. 

#### 3.3.4. *Caloptilia azaleella* (Brants, 1913) San-cheol-jjug-ga-neun-na-bang

*Gracillaria azaleella* Brants, 1913: lxx–lxii [41]. TL: Japan. TD: Unknown. 

*Gracillaria azalea*: Busck, 1914 [42]. 

*Gracillaria anthracosperma*: Meyrick, 1931 [43].

Forewing length 5.1 mm. 

Material examined. [GG] 2♂, Mt. Myeongji, Gapyeong-gun, 30 September 2014 (emergence date: 10 October 2014), gen. slide no. HNUSEL-5611, 5612-coll. UH Heo. 

Distribution. Korea, China, Japan, Russia, Australia, New Zealand, Austria, Belgium, Czech Republic, Denmark, Finland, France, Germany, Hungary, Ireland, Italy, Luxembourg, Netherlands, Norway, Poland, Portugal, Slovakia, Sweden, Switzerland, United Kingdom, Canada, United States, South Africa.

Host plants. *Rhododendron yedoense* var. *poukhanense* in Korea [Ericaceae] [1,44]. *R. decandrum* Makino, *R. indicum* (L.) Sweet, *R. kaempferi* Planch, *R. kiusianum* Makino, *R. macrosepalum* Maxim. *R. viscistylum* Nakai in Japan [Ericaceae] [1,2]. *R.* sp. in New Zealand [Ericaceae] [1,45]. *Azalea* sp. in Austria [Ericaceae] [1,46]. *R. simsii* Planch. in Czech Republic [Ericaceae] [1,47]. *R. japonicum* Sur., *R.* sp. in Denmark [Ericaceae] [1,48,49]. *R. obtusum* (Lindl.) Planch. in France [Ericaceae] [1,50]. *R. indicum* (L.) Sweet, *R. simsii* Planch., *R.* sp. in Germany [Ericaceae] [1,51,52,53,54]. *R. indicum* (L.) Sweet, *R. japonicum* Sur., *R.* sp. in Netherlands [Ericaceae] [1,41,54]. *R.* sp. in Portugal [Ericaceae] [1,55]. *R. indicum* (L.) Sweet in Slovakia [Ericaceae] [1,56]. *R.* sp. in Sweden [Ericaceae] [1,57]. *R. indicum* (L.) Sweet in Switzerland [Ericaceae] [1,58]. *R. indicum* (L.) Sweet, *R. simsii* Planch. *R.* sp. in United Kingdom [Ericaceae] [1,59,60]. *R. indicum* (L.) Sweet, *R.* sp. in United states [Ericaceae] [1,42,61]. *R. indicum* (L.) Sweet in South Africa [Ericaceae] [1,39].

#### 3.3.5. *Caloptiliaceltidis* Kumata, 1982 Huin-mu-nui-ga-neun-na-bang

*Caloptilia celtidis* Kumata, 1982: 76–79 [2]. TL: Honshu, Japan. TD: EIHU (Holotype); EIHU, BMNH (Paratypes).

Forewing length 5 mm.

Material examined. [JN] 2♀, Mt. Wangui, Haeryong-myeon, Suncheon-si. 30 July 2016 (emergence date: 10 August 2016), gen. slide no. HNUSEL-5613-coll. UH Heo; 1♀, experiment forest of Seoul University, Gwangyang-si, 13 September 2016 (emergence date: 29 September 2016)-coll. UH Heo; 1♂, Mt. Jogye, Suncheon-si, 4 July 2018 (leg. BK Byun), gen. slide no. HNUSEL-5561-coll. HNUSEL. 

Distribution. Korea (new record), China, Hong Kong, Japan. 

Host plants. *Celtis sinensis* Persoon [Cannabaceae] in Korea (in this study). *C. sinensis* Persoon [Cannabaceae] in China [1,20]. *C. jessoensis* Koidz., *C. sinensis* Persoon [Cannabaceae] in Japan [1,2].

Remarks. This species was reared from *Celtis sinensis* Persoon of the family Cannabaceae in this study.

#### 3.3.6. *Caloptilia chrysolampra* (Meyrick, 1936) Beo-deul-ga-neun-na-bang

*Gracillaria chrysolampra* Meyrick, 1936: 38 [62]. TL: Taiwan. TD: BMNH (Syntypes). 

*Caloptilia chrysolampra*: Issiki, 1957 [30].

Forewing length 5 mm. 

Material examined. [GJ] 1♀, Pyeong-dong, Gwangsan-gu, 20 May 2018 (leg. BK Byun), gen. slide no. HNUSEL-5584-coll. HNUSEL; 2♀, Pyeong-dong, Gwangsan-gu, 21 May 2018 (leg. BK Byun), gen. slide no. HNUSEL-5659-coll. HNUSEL; [GG] 1♀, Gwang Neung, 8 June 1977 (leg. KT Park)-coll. HNUSEL; 1♀, Gwangrung, 7 August 1986 (leg. KT Park, MK Ko)-coll. HNUSEL; 1♂, Sangnim-ri, Docheok-myeon, Gwangju, 11 August 2018 (leg. BK Byun), gen. slide no. HNUSEL-5658-coll. HNUSEL; [GW] 1♂, Chugok, 30 July 1986 (leg. KT Park, U Park)-coll. HNUSEL; 2♀, Mt. Samak, 19 July 1989 (leg. KT Park), gen. slide no. HNUSEL-5040-coll. HNUSEL; 2♀, Mt. Seolak, 9 August 1989 (leg. BK Byun)-coll. HNUSEL; 1♂, Mt. Jeombong, Girin-myeon, Inje, 3 August 2017 (leg. Lim, Lee, Choi, Shin, Roh)-coll. KNAE. 

Distribution. Korea, China, Japan, Taiwan. 

Host plants. *Salix pseudo-lasiopyne* in Korea [Salicaceae] [1,44]. *Populus nigra* L., *S. babylonica* L., 1753 in China [Salicaceae] [1,20]. *P. nigra* L., *S. babylonica* L., in Japan [Salicaceae] [1,2]. *S.* sp. in Taiwan [Salicaceae] [1,62].

#### 3.3.7. *Caloptilia dentata* Liu & Yuan, 1990 Ne-mo-mu-nui-ga-neun-na-bang

*Caloptilia (Caloptilia) dentata* Liu & Yuan, 1990: 186 [20]. TL: Beijing, China. TD: IZAS (Holotype; Paratypes). 

Forewing length 5 mm. 

Material examined. [JB] 2♂, 1♀, Jangan-ri, Songgwang-myeon, Suncheon-si, 4 July 2018 (leg. BK Byun), gen. slide no. HNUSEL-5504-coll. HNUSEL; 2♀, Seji-myeon, Naju-si, 31 July 2018 (leg. BK Byun)-coll. HNUSEL; [GN] 1♂, Mt. Keum, Namhae, 7 August 1982 (leg. KT Park), gen. slide no. HNUSEL-5473-coll. HNU

Distribution. Korea (new record), China. 

Host plants. *Acer truncatum* Bunge [Sapindaceae] in China [1,20].

#### 3.3.8. *Caloptilia hidakensis* Kumata, 1966 Go-lo-soe-ga-neun-na-bang

*Caloptilia hidakensis* Kumata, 1966: 4 [19]. TL: Hokkaido, Japan. TD: EIHU (Holotype; Paratype). 

Forewing length 6 mm. 

Material examined. [GG] 1♀, Mt. Hwaak, 25 August 1998 (Paek, Lee & Ahn), gen. slide no. HNUSEL-5047-coll. INU; 1♂, Mt. Bongmisan, Danwolmyeon, Yangpyeong, 22 October 2008 (leg. BW Lee, SY Park, SR Kim), gen. slide no. HNUSEL-5391-coll. HNUSEL; [GW] 1♀, Mt. Bangadariyaksuteo, Odaesan, Pyeongchang, 22 July 2009 (leg. SY Park & JS Lim), gen. slide no. HNUSEL-5548-coll. HNUSEL; [JN] 1♀, Suncheon, 4 September 1991 (leg. SS Kim)-coll. HNUSEL. 

Distribution. Korea, Japan, Russia. 

Host plants. *Acer pictum* Thunb. ex Murray [Sapindaceae] in Japan [1,19]. 

#### 3.3.9. *Caloptilia kadsurae* Kumata, 1966 Nam-o-mi-ja-ga-neun-na-bang

*Caloptilia kadsurae* Kumata, 1966: 19 [19]. TL: Honshu, Japan. TD: EIHU (Holotype; Paratypes).

Forewing length 5.2 mm. 

Material examined. [JN] 2♂, Wando Arboretum, Wando, 31 August 2019 (emergence date: 16 September 2019) (leg. UH Heo), gen. slide no. HNUSEL-5593, 5594-coll. UH Heo; [JJ] 1♀, Sanghyo-dong, Seogwipo-si, 24 July 2019 (leg. BK Byun), gen. slide no. HNUSEL-5558-coll. HNUSEL. 

Distribution. Korea (new record), Japan. 

Host plants. *Kadsura japonica* Dunal [Magnoliaceae] in Korea (in this study). *Kadsura japonica* Dunal [Magnoliaceae] in Japan [1,19].

Remarks. This species was reared from *Kadsura japonica* Dunal of the family Magnoliaceae in this study. 

#### 3.3.10. *Caloptilia kisoensis Kumata*, 1982 Sin-na-mu-ga-neun-na-bang

*Caloptilia (Caloptilia) kisoensis* Kumata, 1982: 45–47 [2]. TL: Honshu, Japan. TD: EIHU (Holotype; Paratypes). 

Forewing length 5.1 mm. 

Material examined. [DJ] 1♀, Mt. Manin, Haso-dong, Dong-gu, 26 May 2018 (leg. BK Byun), gen. slide no. HNUSEL-5578-coll. HNUSEL; [GG] 1♀, Mt. Suri-san, Kunpo City, 25 July 1997 (leg. JS Lee & HK Lee)-coll. HNUSEL; [GW] 1♀, Chuncheon-Dam, 15 June 1995 (leg. HK Lee & MS Go)-coll. HNUSEL; 1♀, Mt. Taegisan, Dunnae-myeon, Hoengseong-gun, 5 June 2018 (leg. TG Lee, HK Kim, CM Jang), gen. slide no. HNUSEL-5660-coll. INU. 

Distribution. Korea, Japan, Russia. 

Host plants. *Acer ginnala* Maxim., *A. pictum* Thunb. ex Murray [Sapindaceae] in Korea [1,41]. *A. ginnala* Maxim., *A. pictum* Thunb. ex Murray [Sapindaceae] in Japan [1,2].

#### 3.3.11. *Caloptilia koreana* sp. nov. Kim and Byun, 2022 Han-gug-ga-neun-na-bang

Type. Holotype. [KOREA] [GG] 1♀, Lake Yuklim, Gwangneung, 7 September 2017 (leg. Lim, Kim, Lee, Shin, Roh), gen. slide no. HNUSEL-5575-coll. KNAE.

Distribution. Korea (endemic, Gwangneung). 

Host plants. Unknown. 

#### 3.3.12. *Caloptilia leucothoes* Kumata, 1982 San-jin-dal-lae-ga-neun-na-bang

*Caloptilia (Caloptilia) leucothoes* Kumata, 1982: 68 [2]. TL: Hokkaido, Japan. TD: EIHU (Holotype), EIHU, BMNH (Paratypes).

Forewing length 4.3 mm.

Material examined. [DJ] 1♂, Daedeok, Jeonming-dong, 5 June 2015 (leg. BK Byun), gen. slide no. HNUSEL-5394-coll. HNUSEL; [GG] 1♀, Yeoncheon, Mt. Godae, 30 April 2001 (leg. Bae, Kim & Kim)-coll. INU; 1♀, Obsevatory, Korea National Arboretum, Gwangneung, 16 July 2018 (leg. BS Park, YM Shin)-coll. KNAE; [GW] 1♀, Chuncheon, 17 June 1984 (leg. KT Park)-coll. HNUSEL; 2♂, Chuncheon, 21 March 1993 (leg. BK Byun), gen. slide no. HNUSEL-5547-coll. HNUSEL; 1♀, Mt. Taegisan, Dunnae-myeon, Hoengseong-gun, 5 June 2018 (leg. TG Lee, HK Kim, CM Jang), gen. slide no. HNUSEL-5564-coll. INU. 

Distribution. Korea, Japan, Russia.

Host plants. *Leucothoe grayana* Maxim., *Rhododendron albrechti* Maxim., *R. dauricum* L., *R. dilatatum* Miq., *R. reticulatum* D. Don, *R.* sp. [Ericaceae] in Japan [1,2]. *Menziesia pentandra* Maxim. [Ericaceae] in Russia [1,29].

#### 3.3.13. *Caloptilia magnoliae* Kumata, 1966 Mog-lyeon-ga-neun-na-bang

*Caloptilia magnoliae* Kumata, 1966: 17 [19]. TL: Hokkaido, Japan. TD: EIHU (Holotype; Paratypes). 

Forewing length 8 mm. 

Material examined. [GG] 1♂, Mt. Myeongji, Gapyeong-gun, 8 September 2013 (emergence date: 15 September 2013) (leg. UH Heo), gen. slide no. HNUSEL-5609, 5661-coll. HNUSEL; [JB] 1♀, Mt. Degyu, 22 July 1983 (leg. SB An)-coll. HNUSEL. 

Distribution. Korea, Japan. 

Host plants. *Machilus thunbergii* Siebold & Zucc. [Lauraceae] (in this study); *Magnolia* sp. [Magnoliaceae] in Korea [1,63]. *M. kobus* DC [Magnoliaceae] in Japan [1,19].

#### 3.3.14. *Caloptilia mandschurica* (Christoph, 1882) Bug-bang-min-ga-neun-na-bang

*Gracillaria mandschurica* Christoph, 1882: 39–40 [64]. TL: Russia. TD: BMNH (Holotype; Allotype). 

*Caloptilia (Caloptilia) mongolicae*: Kumata, 1982 [2]. 

Forewing length 6 mm. 

Material examined. [SU] 3♂, 2♀, Hongneung, 1 April 1998 (leg. BK Byun) gen. slide no. HNUSEL-5388, 5393-coll. HNUSEL; [DJ] Daedeok, Jeonming-dong, 13 March 2013 (leg. BK Byun)-coll. HNUSEL; [SJ] 3♂, 8♂, Sejong-si, Geumnam, 30 April 2015 (leg. BK Byun), gen. slide no. HNUSEL-5545, 5546-coll. HNUSEL; [GG] 1♂, Suweon, 13 April 1977 (leg. KT Park)-coll. HNUSEL; [GW] 1♂, Chuncheon 11 June 1989 (leg. KT Park), gen. slide no. HNUSEL-5054-coll. HNUSEL; 1♂, Chuncheon, 27 July 1985 (leg. KT Park)-coll. HNUSEL; 1♀, Sambae-ri, Gonggeoun-myeon, Hoenseong-gun, 27 September 2016 (leg. SM Na, DJ Lee)-coll. INU; 2♀, Mt. Taegisan, Dunnae-myeon, Heongseong-gun, 5 June 2018 (leg. TG Lee, HK Kim, CM Jang)-coll. INU; 1♀, Mt. Jeombong, Gilin-myeon, Inge-gun, 25 July 2018 (BS Park, YM Shin, SB Choi)-coll. KNAE; [CB] 2♂, Boeun, Songnisan-myeon, Galmok-ri, 5 May 2017 (leg. BK Byun)-coll. HNUSEL; [CN] 1♂, Daegok-ri, Haemi-myeon, Seosan-si, 16 May 2019 (leg. BK Byun)-coll. HNUSEL; [JB] 1♀, Cheoncheon-myeon, Wolgok-ri, 5 April 2013 (leg. BK Byun)-coll. HNUSEL; [JN] 3♀, Mt. Jogye, Suncheon-si, 4 July 2018 (leg. BK Byun)-coll. HNUSEL. 

Distribution. Korea, China, Japan, Russia.

Host plants. *Quercus mongolica* subsp. *crispula* (Blume) [Fagaceae] in Korea (in this study). *Castanea crenata* Siebold & Zucc., *Q. acutissima* Carruth., *Q. dentata* Thunb., *Q. mongolica* subsp. crispula (Blume) Menitsky in China [Fagaceae] [1,20]. *C. crenata* Siebold & Zucc., *Q. acutissima* Carruth., *Q. crispula* Blume, *Q. dentata* Thunb., *Q. mongolica* subsp. *crispula* (Blume) Menitsky, *Q. serrata* Thunb. in Japan [Fagaceae] [1,2,65,66]. *Q. mongolica* Fisch. & Turcz. in Russia [Fagaceae] [1,64].

Remarks. This species was reared from *Quercus mongolica* subsp. *crispula* (Blume) of the family Fagaceae in this study.

#### 3.3.15. *Caloptilia monticola* Kumata, 1966 Oe-ban-jeom-ga-neun-na-bang

*Caloptilia monticola* Kumata, 1966: 8 [19]. TL: Honshu, Japan. TD: EIHU (Holotype; Paratypes). 

Forewing length 6.21 mm

Material examined. [GB] 1♀, Mt. Tonggo, Geumgangsong-myeon, Uljin-gun, 23 June 2017 (leg. BS Park, SM Na, DJ Lee), gen. slide no. HNUSEL-5587-coll. INU. 

Distribution. Korea (new record), China, Japan, Russia. 

Host plants. *Acer argutum* Maxim., *A. ginnala* Maxim., *A. pentaphyllum* Diels., *A. pictum* Thunb. ex Murray, *A. rufinerve* Siebold & Zucc., *A. semenovii* Regel & Herd, *A. ukurunduense* Trautv. & Mey. [Sapindaceae] in China [1,20,27]. *A. argutum* Maxim., *A. distylum* Siebold & Zucc., *A. ginnala* Maxim., *A. micranthum* Siebold & Zucc., *A. pictum* Thunb. ex Murray, *A. rufinerve* Siebold & Zucc., *A. tschonoskii* Maxim., *A. ukurunduense* Trautv. & Mey. [Sapindaceae] in Japan [1,2,19]. *A. pictum* Thunb. ex Murray, *A. semenovii* Regel & Herd, *A.* sp., *A. ukurunduense* Trautv. & Mey. [Sapindaceae] in Russia [1,28,29].

#### 3.3.16. *Caloptilia pulverea* Kumata, 1966 Heug-gal-saeg-jeom-ga-neun-na-bang

*Caloptilia pulverea* Kumata, 1966:13 [19]. TL: Hokkaido, Japan. TD: EIHU (Holotype), EIHU, BMNH (Paratypes). 

Forewing length 6 mm.

Material examined. [GN] 1♂, 1♀, Mt. Gaji, 19 August 1993 (leg. KT Park & BK Byun), gen. slide no. HNUSEL-5055, 5605-coll. HNUSEL. 

Distribution. Korea, China, Japan, Russia. 

Host plants. *Alnus hirsuta* Turcz., *A. japonica* (Thunb.) Steud., *A. matsumurae* Callier, *A. maximowiczii* Callier, *A. rugosa* (Du Roi) Spreng. in China [Betulaceae] [1,20]. *A. firma* Siebold & Zucc., *A. japonica* (Thunb.) Steud., A. matsumurae Callier, A. maximowiczii Callier, A. serratuloides Callier in Japan [Betulaceae] [1,2,19]. *A. japonica* (Thunb.) Steud., *A. maximowiczii* Callier in Russia [Betulaceae] [1,28].

#### 3.3.17. *Caloptilia purpureus* sp. nov. Kim and Byun, 2022 Bo-la-ga-neun-na-bang

Type. Holotype. [KOREA] [JN] 1♂, Mt. Geumo, Dolsan-eup, Yeosu-si, 10 October 2018 (emergence data: 28 October 2018) (leg. UH Heo), gen. slide no. HNUSEL-5592-coll. UH Heo; Paratype. [JN] 1♀, Mt. Geumo, Dolsan-eup, Yeosu-si, 10 October 2018 (emergence data: 1 November 2018) (leg. UH Heo), gen. slide no. HNUSEL-5591-coll. UH Heo.

Distribution. Korea (endemic, Yeosu-si). 

Host plants. *Sageretia theezans* Brongn. [Rhamnaceae] in Korea (in this study).

Remarks. This species was reared from *S. theezans* Brongn. of the family Rhamnaceae in this study. 

#### 3.3.18. *Caloptilia pyrrhaspis* (Meyrick, 1931) No-lang-jul-ga-neun-na-bang

*Gracillaria pyrrhaspis* Meyrick, 1931: 17 [43]. TL: Szechuan, China. TD: BMNH (Holotype). 

*Caloptilia bicolor* Ermolaev, 1977: 105, 110 [67]. 

*Caloptilia pyrrhaspis*: Kumata, 1982: 74 [2]. 

Forewing length 4 mm.

Material examined. [GG] 1♀, Mt. Myeongji, Gapyeong-gun, 8 September 2013 (emergence date: 25 September 2013) (leg. UH Heo), gen. slide no. HNUSEL-5596-coll. HNUSEL; [GB] 1♂, Bonghwa-gun, Seokpo-ri, 14 August 2013 (leg. BK Byun), gen. slide no. HNUSEL-5570-coll. HNUSEL. 

Distribution. Korea, China, Japan, Russia.

Host plants. *Betula davurica* Pall. [Betulaceae] in Korea (in this study). *Betula dahurica* Pall., *B. ermanii* Chamisso, *B. grossa* Siebold & Zucc. [Betulaceae] in Chian [1,20]. *B. ermanii* Chamisso, *B. grossa* Siebold & Zucc. [Betulaceae] in Japan [1,2]. *B. dahurica* Pall., *B. platyphylla* Sukaczev [Betulaceae] in Russia [1,29,67].

Remarks. This species was reared from Betula davurica Pall. of the family Betulaceae in this study.

#### 3.3.19. *Caloptilia recitata* (Meyrick, 1918) Keun-gal-saeg-mu-nui-ga-neun-na-bang 

*Gracillaria recitata* Meyrick, 1918b: 178–179 [21]. TL: Assam, India. TD: BMNH (Syntypes). 

*Caloptilia recitata*: Issiki, 1957: 30 [30].

Forewing length 6 mm.

Material examined. [DJ] 1♂, Daedeok, Jeonmin-dong, 5 June 2015 (leg. BK Byun), gen. slide no. HNUSEL-5392-coll. HNUSEL; [JN] 1♂, Chusan-ri, Ongryong-myeon, Gwangyang, 2 June 2018 (leg. BK Byun), gen. slide no. HNUSEL-5560-coll. HNUSEL; [JB] 4♂, 4♀, Sangseo-myeon, Buan-gun, 29 September 2018 (leg. BK Byun), gen. slide no. HNUSEL-5543, 5544-coll. HNUSEL. 

Distribution. Korea (new record), China, Hong Kong, Japan, India, Nepal.

Host plants. *Cotinus coggygria* Scop., *Rhus javanica* L., *Toxicodendron sylvestre* (Siebold & Zucc.) Kuntze, *T. trichocarpum* (Miq.) Kuntze in China [Anacardiaceae] [1,20]. *R. javanica* L., *T. sylvestre* (Siebold & Zucc.) Kuntze, *T. trichocarpum* (Miq.) Kuntze in Japan [Anacardiaceae] [1,2]. *R. javanica* L. in Nepal [Anacardiaceae] [1,2].

Remarks. This species was collected from *Ailanthus altissima* (Mill.) [Simaroubaceae] with pupal cocoon on the back side.

#### 3.3.20. *Caloptilia rhois* Kumata, 1982 Ot-na-mu-ga-neun-na-bang 

*Caloptilia (Caloptilia) rhois* Kumata, 1982: 62–65 [2]. TL: Honshu, Japan. TD: EIHU (Holotype; Paratypes).

Forewing length 6 mm.

Material examined. [SJ] 1♂, Sejong-si, Geumnam, 30 April 2015 (leg. BK Byun) gen. slide no. HNUSEL-5555-coll. HNUSEL; [DJ] 1♂, Daedeok, Jeonmin-dong, 9 April 2014 (leg. BK Byun) gen. slide no. HNUSEL-5588-coll. HNUSEL; [JB] 1♂, Eumnae-ri, Mugu-eup, Muju-gun, 9 August 2019 (leg. BK Byun) gen. slide no. HNUSEL-5586-coll. HNUSEL; [JN] 2♂, Mt. Gahak, 26 July 2005 (leg. BK Byun) gen. slide no. HNUSEL-4156, 5060-coll. HNUSEL. 

Distribution. Korea, China, Japan, Hong Kong. 

Host plants. *Rhus javanica* L., *Toxicodendron succedaneum* (L.) Kuntze in China [Anacardiaceae] [1,24]. *R. javanica* L., *T. succedaneum* (L.) Kuntze in Japan [Anacardiaceae] [1,2].

#### 3.3.21. *Caloptilia sapporella* (Matsumura, 1931) Jol-cham-na-mu-ga-neun-na-bang 

*Gracillaria sapporella* Matsumura, 1931: 1101 [68]. TL: Japan. TD: EIHU (Syntype). 

*Caloptilia illicii* Kumata, 1966: Shin et al., 2015 (misidentification) [5].

Forewing length 5 mm. 

Material examined. [SJ] 1♂, Geumcheon-ri, Geumnam-myeon, 17 August 2017 (leg. SM Na, YB Cha), gen. slide no. HNUSEL-5583-coll. INU; [GG] 1♂, Gwangleung, 28 July 2000 (leg. BK Byun, WI Bae), gen. slide no. HNUSEL-5573-coll. HNUSEL; 1♀, Sangsaek-ri, Gapyeong-eup, Gapyeong-gun (leg. BK Byun), gen. slide no. HNUSEL-5568-coll. HNUSEL; 1♂, Sohol-eup, Pocheon-si, 13 August 2019 (leg. BK Byun), gen. slide no. HNUSEL-5572-coll. HNUSEL; [GW] 1♂, Janggoal, Chuncheon, 22 July 1995 (leg. JS Lee), gen. slide no. HNUSEL-5562-coll. HNUSEL; 1♀, Bukbang-myeon, Hongcheon-gun, 26 August 2019 (leg. BK Byun), gen. slide no. HNUSEL-5585-coll. HNUSEL; [JN] 1♂, Mt. Wolchul, Seongjeon, Gangjin, 5 May 1999 (leg. HG Lee), gen. slide no. 5582-coll. NAAS; 1♂, Mt. Baekun, Gwangyang, 18 May 2012 (leg. BK Byun), gen. slide no. HNUSEL-5037-coll. HNUSEL; 1♀, Chusan-ri, Ongryong-myeong, Gwangyang, 28 May 2018 (leg. BK Byun), gen. slide no. HNUSEL-5567-coll. HNUSEL; [GN] 1♂, Namhae, 1 June 1994 (leg. BK Byun), gen. slide no. NAK-534-coll. HNUSEL; 1♂, same locality, 2 June 1994 (leg. BK Byun), gen. slide no. NAK-535-coll. HNUSEL. 

Distribution. Korea, China, Japan, Russia. 

Host plants. *Quercus acutissima* Carruth. in Korea [Fagaceae] [1,63]. *Castanea crenata* Siebold & Zucc., *Q. acuminata* (Michx.) Sargent, *Q. dentata* Thunb., *Q. mongolica* subsp. *crispula* (Blume) Menitsky, *Q. serrata* Thunb. in China [Fagaceae] [1,20]. *C. crenata* Siebold & Zucc., *Q. acutissima* Carruth., *Q. crispula* Blume, *Q. dentata* Thunb., *Q. mongolica* subsp. *crispula* (Blume) Menitsky, *Q. serrata* Thunb. in Japan [Fagaceae] [1,2,66]. *Q. dentata* Thunb., *Q. mongolica* Fisch. & Turcz., *Q. mongolica* subsp. *crispula* (Blume) Menitsky in Russia [Fagaceae] [1,29,69]. 

Remarks. This species was misidentified in Shin et al. 2015, as *Caloptilia sapporella* to *C. fidella*. 

#### 3.3.22. *Caloptilia schisandrae* Kumata, 1966 O-mi-ja-ga-neun-na-bang 

*Caloptilia schisandrae* Kumata, 1966: 18 [19]. TL: Hokkaido, Japan. TD: EIHU (Holotype; Paratypes). 

Forewing length 7 mm. 

Material examined. [GW] 1♂, Mt. Gariwang, Jeongseon, 30 July 2013 (leg. SS Kim), gen. slide no. HNUSEL-5389-coll. HNUSEL. 

Distribution. Korea, China, Japan, Russia. 

Host plants. *Schisandra chinensis* Baill. in Korea [Magnoliaceae] [1,70]. *S. chinensis* Baill. in Japan [Magnoliaceae] [1,19]. *S. chinensis* Baill. in Russia [Magnoliaceae] [1,31].

#### 3.3.23. *Caloptilia soyella* (van Deventer, 1904) Tob-ni-ga-neun-na-bang 

*Gracillaria soyella* van Deventer, 1904: 22–25 [22]. TL: Java, Indonesia. TD: RNHL (Lectotype; Paralectotype). 

*Caloptilia soyella*: Issiki, 1950: 451 [31].

Forewing length 4.2 mm. 

Material examined. [JN] 1♀, Mt. Jeseok, Boseong-gun, 15 September 2016 (emergence date: 29 September 2016), (leg. UH Heo), gen. slide no. HNUSEL-5597-coll. UH Heo; 2♂, Seji-myeon, Naju-si, 31 July 2018 (leg. BK Byun), gen. slide no. HNUSEL-5556, 5557-coll. HNUSEL. 

Distribution. Korea (new record), China, Japan, India, Indonesia, Viet Nam, Fiji, Cape Verde. 

Host plants. *Lespedeza cyrtobotrya* Miq. [Fabaceae] in Korea (in this study). *Cajanus cajan* (L.) Millsp., *Glycine max* (L.) Merr., *Kummerovia striana* Schindler, *Lespedeza cyrtobotrya* Miq., *Phaseolus mungo* L., *Vigna angularis* (Willd.) Ohwi & H. Ohashi in China [Fabaceae] [1,20]. *G. max* (L.) Merr., *K. striana* Schindler, *L. cyrtobotrya* Miq., *P. calcaratus* Roxb., *V. angularis* (Willd.) Ohwi & H. Ohashi in Japan [Fabaceae] [1,2]. *C. cajan* (L.) Millsp., *P. mungo* L. in India [Fabaceae] [1,32]. *Soya hispida* Moench in Indonesia [Fabaceae] [1,22].

Remarks. This species was reared from *Lespedeza cyrtobotrya* Miq. of the family Fabaceae in this study. 

#### 3.3.24. *Caloptilia stigmatella* (Fabricius, 1781) Baeg-yang-na-mu-ga-neun-na-bang 

*Tinea stigmatella* Fabricius, 1781: 295–296 [71]. TL: United Kingdom. TD: GLAHM (Holotype). 

*Gracillaria consimilella* Frey & Boll, 1876 [72]. 

*Phalaena cruciella* Goeze, 1783 [73].

*Tinea equestris* de Fourcroy, 1785 [74]. 

*Gracillaria ochracea* Haworth, 1828 [75].

*Gracillaria purpurea* Haworth, 1828 [75]. 

*Gracillaria purpuriella* Chambers, 1872 [76].

*Tinea triangulella* Panzer, 1794 [77].

*Tinea triangolosella* Costa, 1836 [78]. 

*Gracillaria trigona* Haworth, 1828 [75].

*Tinea upupaepennella* Hübner, 1796 [34]. 

Forewing length 6–6.2 mm.

Material examined. [GW] 1♂, 1♀, Sogumgang, 8 August 1988 (leg. KT Park), gen. slide no. HNUSEL-5041, 5043-coll. HNUSEL; [GB] 1♀, Bonghwa-gun, Seokpo-ri, 14 August 2013 (leg. BK Byun)-coll. HNUSEL. 

Distribution. Korea, China, Japan, Russia, Mongolia, India, Armenia, Austria, Belgium, Bosnia and Herzegovina, Bulgaria, Croatia, Czech Republic, Estonia, Finland, France, Georgia, Germany, Hungary, Ireland, Italy, Kyrgyzstan, Latvia, Leichtenstein, Lithuania, Luxembourg, Macedonia, Morocco, Netherlands, Norway, Poland, Portugal, Romania, Serbia, Slovakia, Spain, Sweden, Switzerland, Tajikstan, Turkey, Turkmenistan, Ukraine, Uzbekistan, United Kingdom, Canada, United states

Host plants. *Populus* sp. in Korea [*Salicaceae*] [1,63]. *P. nigra* L., *Salix bakko* Kimura, *Salix miyabeana* (Seemen), *S.* sp. in Japan [*Salicaceae*] [1,2,79]. *Robinia pseudacacia* L. in Russia [Fabaceae] [1,80]; *Myrica gale* L., in Russia [Myricaceae] [1,80]; *Chosenia arbutifolia* (Pall.) A. K. Skvortsov, *P. nigra* L., P. suaveolens Fish. ex Loud., *P. tremula* L., *P.* sp., *S. caprea* L., *S. sachalinensis* Schmidt & Sekka, *S.* sp. in Russia [*Salicaceae*] [1,28,29,81,82]. *P. nigra* var. italica Münchh., *P.* sp. *S. alba* L., *S. eleagnos* Scop., *S. purpurea* L., *S. repens* L. *S.* sp. in Austria [*Salicaceae*] [83,84,85]. *P.* sp., *S. repens* L. in Belgium [*Salicaceae*] [1,86]. *S. purpurea* L. in Bulgaria [*Salicaceae*] [1,87]. *P. alba* L., *P. tremula* L., *S. cinerea* L., *S. incanaelaeagnos* Scop. in Czech Republic [*Salicaceae*] [88,89]. *P. alba* L., *P.* sp., *S.* incanaelaeagnos Scop., *S. lanata* L., *S.* sp. in Germany [*Salicaceae*] [1,51,90,91,92]. *P. alba* L., *P. tremula* L., *S. incanaelaeagnos* Scop., *S.* sp. in Italy [*Salicaceae*] [1,93,94,95]. *P. tremula* L. in Latvia [*Salicaceae*] [1,96]. *P.* sp., *P. tremula* L., *S.* sp. in Lithuania [1,97,98]. *S. pedicellata* Desf. in Morocco [*Salicaceae*] [99]. *P. nigra* var. *italica* Münchh., *P. tremula* L., *S. triandra* L., *Salix* sp. in Netherlands [*Salicaceae*] [1,100,101]. *P.* sp., *S.* sp. in Norway [*Salicaceae*] [1,102]. *P. alba* L., *P. tremula* L., *P.* × *canadensis* Moench., *S. alba* L., *S. caprea* L., *S. fragilis* L., *S. purpurea* L., *S. triandra* L. in Poland [*Salicaceae*] [1,103,104,105]. *P. alba* L., *S. atrocinerea* Brot., *S. salviifolia* Brot., *S.* × *fragilis* L. in Portugal [*Salicaceae*] [1,106]. *P. nigra* var. *italica* Münchh., *P.* sp., *S. caprea* L., *S.* sp. in Slovakia [*Salicaceae*] [1,90,107]. *P.* sp., *S.* sp. in Sweden [*Salicaceae*] [1,57]. *P. nigra* var. *italica* Münchh., *P. tremula* L., *S.* sp. in Switzerland [*Salicaceae*] [1,108]. *P.* sp., *S.* sp. [*Salicaceae*] in United Kingdom [1,59]. *P.* sp., *S. longifolia* Lam., *S.* sp. in United states [1,76,109].

#### 3.3.25. *Caloptilia syrphetias* (Meyrick, 1907) Hu-bag-na-mu-ga-neun-na-bang 

*Gracilaria syrphetias* Meyrick, 1907: 984 [110]. TL: Ceylon, Sri Lanka. TD: NHMUK.

*Gracilaria zopherotarsa*: Meyrick, 1936: 39 [62].

*Caloptilia perseella*: Kumata, 1982: 93 [2].

Forewing length 7.1 mm.

Distribution. Korea (Jeju Island), Brunei Darussalam, China, Hong Kong, India, Indonesia, Japan, Malaysia, Sri Lanka, Thailand.

Host plants. *Machilus thunbergii* Siebold et Zucc. [Lauraceae] [16].

#### 3.3.26. *Caloptilia theivora* (Walsingham, 1891) Dong-baeg-ga-neun-na-bang 

*Gracillaria theivora* Walsingham, 1891: 49–50 [111]. TL: Ceylon, Sri Lanka. TD: BMNH (Lectotype; Paralectotype).

*Caloptilia theivora*: Issiki, 1950 [31].

Forewing length 5 mm.

Material examined. [GB] 1♀, Uleung-gun, Naegujeon, 15 July 2014 (leg. BK Byun), gen. slide no. HNUSEL-5600-coll. HNUSEL; [JN] 1♂, Suncheon, 4 November 1991 (leg. SS Kim); 1♂, Suncheon, 6 November 1991 (leg. SS Kim), gen. slide no. HNUSEL-5061-coll. HNUSEL; 1♂, Boseong, Hoeryeong-ri, 8 October 2014 (leg. BK Byun), gen. slide no. HNUSEL-5657-coll. HNUSEL; 1♀, Mt. Jogey, Suncheon-si, 4 July 2018 (leg. BK Byun); Ongnyong-myeon, Gwangyang-si, 8 August 2018 (leg. BK Byun); [JJ] 1♀, Ibseokdong, 30 June 1994 (leg. BK Byun); 1♀, Sanghyo-dong, Seogwipo-si, 1 August 2018 (leg. BK Byun)-coll. HNUSEL. 

Distribution. Korea, China, Hong Kong, Japan, Brunei Darussalam, India, Indonesia, Malaysia, Sri Lanka, Taiwan, Thailand, Viet Nam. 

Host plants. *Camellia sinensis* L. [*Theaceae*] in Korea (in this study). *C. japonica* L., *C. sasanqua* Thunb., *Thea sinensis* L. [*Theaceae*] in China [1,20]. *C. japonica* L., *C. sasanqua* Thunb., *T. sinensis* L. [*Theaceae*] in Japan [1,2]. *C. theifera* Griff., *T. sinensis* L. [*Theaceae*] in India [1,32,111]. *C. theifera* Griff. [*Theaceae*] in Sri Lanka [1,112].

Remarks. This species was reared from *Camellia sinensis* L. of the family *Theaceae* in this study.

#### 3.3.27. *Caloptilia xanthos* sp. nov. Kim and Byun, 2022 No-rang-jeom-ga-neun-na-bang

Type. Holotype. [KOREA] [JN] 1♀, Namseong-ri, Cheonggye-myeon, Muan-gun, 3 July 2019 (leg. BK Byun), gen. slide no. HNUSEL-5602-coll. HNUSEL.

Distribution. Korea (endemic, Muan-gun). 

Host plants. Unknown.

#### 3.3.28. *Caloptilia yasudai* Kumata, 1982 No-lan-i-ppal-ga-neun-na-bang 

*Caloptilia yasudai* Kumata, 1982: 51–53 [2]. TL: Hokkaido, Japan. TD: EIHU (Holotype; Paratyeps). 

Forewing length 6 mm.

Distribution. Korea, Japan.

Host plants. Unknown.

Remarks. This species could not be examined in this study.

#### 3.3.29. *Caloptilia zachrysa* (Meyrick, 1907) Sa-gwa-ip-ga-neun-na-bang 

*Gracillaria zachrysa* Meyrick, 1907: 983 [110]. TL: Ceylon, Sri Lanka. TD: BMNH (Syntypes). 

*Caloptilia zachrysa*: Issiki, 1957 [30].

Forewing length 6 mm. 

Material examined. [GG] 1♂, Suweon, 10 July 1976 (leg. KT Park), gen. slide no. HNUSEL-5042-coll. HNUSEL; 1♂, Suweon, 24 September 1980 (leg. CG Yoo), gen. slide no. HNUSEL-5599-coll. HNUSEL; 1♂, Mt. Yeogi, 19 August 1983 (leg. DJ Im)-coll. HNUSEL; 1♂, Gwangleung, 10 July 1990 (leg. KT Park)-coll. HNUSEL. 

Distribution. Korea, China, Japan, India, Sri Lanka, Taiwan. 

Host plants. *Malus pumila* Mill., *Prunus persica* (L.) Batsch in Korea [*Rosaceae*] [1,113]. *M. pumila* Mill., *Photinia glabra* (Thunb.) Maxim., *Rubus* sp. in China [*Rosaceae*] [1,20]. *M. pumila* Mill., *Ph. glabra* (Thunb.) Maxim., *R.* sp. in Japan [*Rosaceae*] [1,2]. *M. pumila* Mill., *M. sylvestris* Mill. in India [*Rosaceae*] [1,21,114,115]. *Ph.* sp. in Taiwan [*Rosaceae*] [1,2].

## 4. Discussion

The genus *Caloptilia* has not been studied extensively in Korea to date, when compared with that in the neighboring countries, especially regarding the number of known species, e.g., Japan has more than 50 reported species [1]. The first record of *Caloptilia* in Korea, which included five species, was reported in 1983 by Park. Later, Park and Han (1986) reported six species of Gracillaridae. In the 2000s and the 2010s, Park and Lee (2001) [70] and Sohn (2007) [44] reported one and four species, respectively. Shin et al. (2015) [5] enumerated 19 species of the genus, with four newly recorded species from Korea. All the known species from Korea including this study, were enumerated with synonymies, distribution and host plant respectively (Appendix A).

The genus *Caloptilia* can be distinguished from the other genera in the family Gracillariidae, based on their males having weakly membranous 7th and 8th abdominal segments, and females with signa present in the corpus bursae of their genitalia. Among the species of Caloptilia, the identification keys can be used to identify each species; however, slight differences exist. In addition, most species of *Caloptilia* morphologically resemble each other, with triangular patches on the forewing, a pair of signa in the female genitalia, and valva dilated toward the apex in the male genitalia. The host plant might be an exact identification key for Gracillariidae moths, because they have high host plant specificity (Davis, 1987 [116]; Brito et al., 2016 [117]), except for some polyphagous species. In this study, we reared a total of nine species of *Caloptilia* from host plants when they were in larval stages. Among them, three species that were investigated provided new host plant information, and one of them, *Ailanthus altissima* (Mill.) was recorded as a new host plant of the family *Gracillariidae*. Based on the host plant survey in this study, all the known host plants were summarized for the gracillariid species (Appendix A). In total 106 species of 15 families were investigated for host plants of *Caloptilia* in the world. Most of them consisted of *Salicaceae* (27 species), *Sapindaceae* (23 species), *Ericaceae* (14 species), *Betulaceae* (12 species) and *Fagaceae* (11 species) [1]. The monophagous character is an obvious characteristic in the genus Caloptilia, but some species were shown to be polyphagous in this study. *Caloptilia stigmatella* were investigated with three families of host plants, and both *C. recitata* and *C. magnoliae* had two families, respectively. 

In future research in Korea, DNA barcodes will function as exact identification keys for microlepidoptera. In this study, we tried to extract the DNA barcodes targeting all species of *Caloptilia* in Korea. However, it was difficult to get successful results, especially on some old specimens or other specimens, due to unknown reasons. We are now preparing and collecting more fresh material. After this, we will try to study the taxonomies and conduct systematic study with species delimitation and DNA data in future. 

## Figures and Tables

**Figure 1 insects-13-01107-f001:**
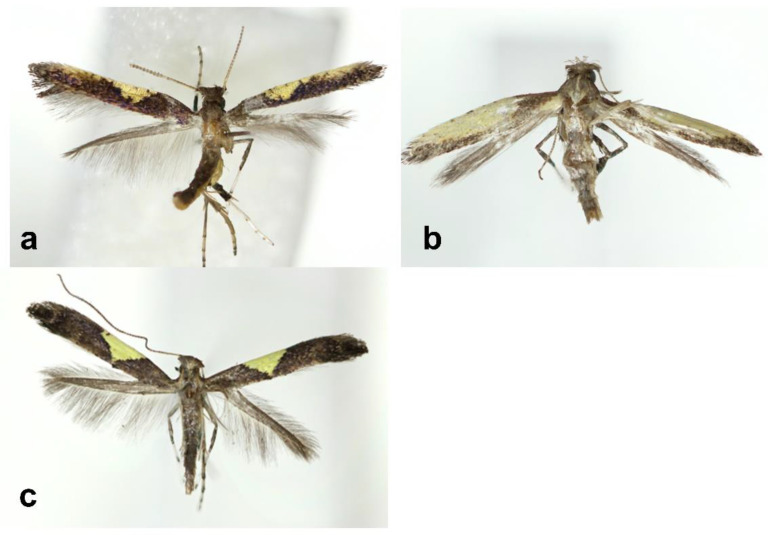
Adults of *Caloptilia*. (**a**). *C. purpureus* (Holotype, gen. slide no. HNUSEL-5592); (**b**). *C. koreana* (Holotype, gen. slide no. HNUSEL-5575); (**c**). *C. xanthos* (Holotype, gen. slide no. HNUSEL-5602).

**Figure 2 insects-13-01107-f002:**
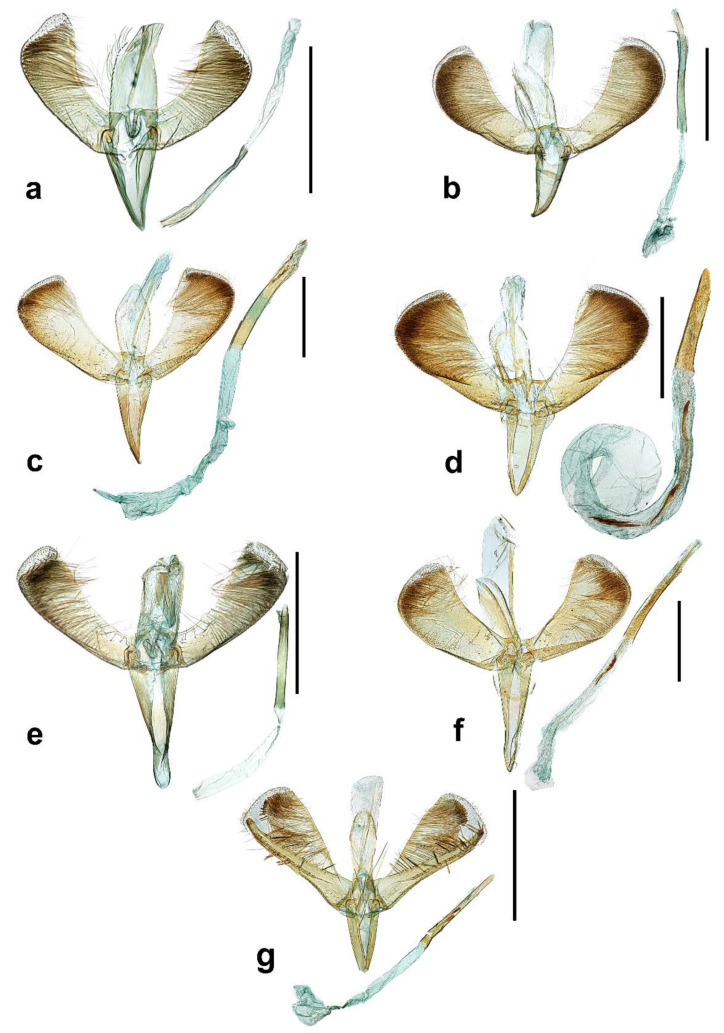
Male genitalia of *Caloptilia*. (**a**). *C. purpureus* (gen. slide no. HNUSEL-5592); (**b**). *C. acericola* (gen. slide no. HNUSEL-5571); (**c**). *C. celtidis* (gen. slide no. HNUSEL-5561); (**d**). *C. dentata* (gen. slide no. HNUSEL-5549); (**e**). *C. kadsurae* (gen. slide no. HNUSEL-5593); (**f**). *C. recitata* (gen. slide no. HNUSEL-5544); (**g**). *C. soyella* (gen. slide no. HNUSEL-5556, 5557) (Scale bars: a–g = 5 mm).

**Figure 3 insects-13-01107-f003:**
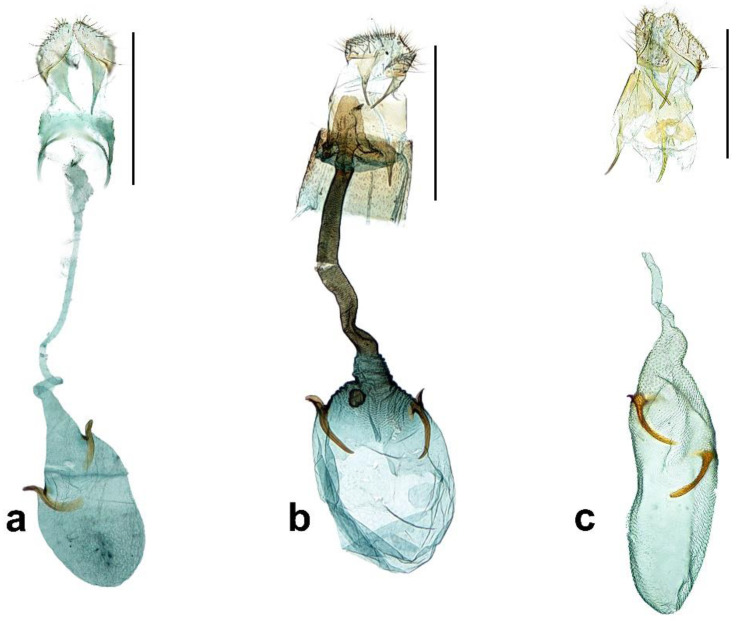
Female genitalia of *Caloptilia*. (**a**). *C. purpureus* (gen. slide no. HNUSEL-5591); (**b**). *C. koreana* (gen. slide no. HNUSEL-5575); (**c**). *C. xanthos* (gen. slide no. HNUSEL-5602). (Scale bars: a–e = 5 mm).

**Figure 4 insects-13-01107-f004:**
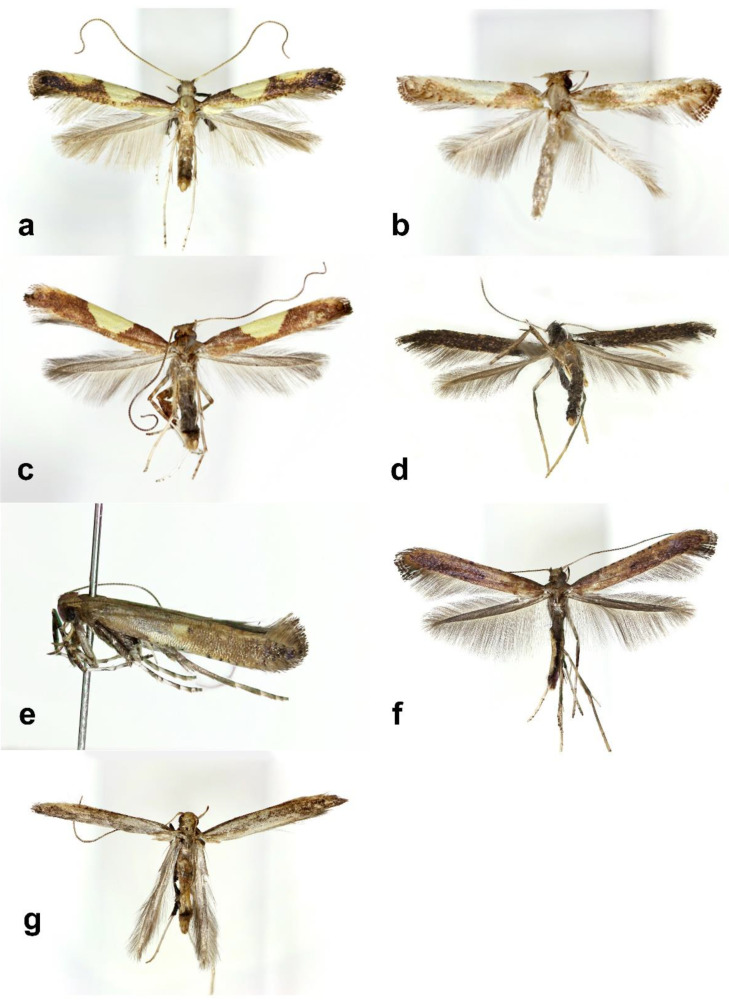
Adults of *Caloptilia*. (**a**). *C. acericola* (gen. slide no. HNUSEL-5577); (**b**). *C. celtidis* (gen. slide no. HNUSEL-5613); (**c**). *C. dentata* (gen. slide no. HNUSEL-5589); (**d**). *C. kadsurae* (gen. slide no. HNUSEL-5593); (**e**). *C. monticola* (gen. slide no. HNU-SEL-5587); (**f**). C. recitata (gen. slide no. HNUSEL-5577); (**g**). *C. soyella* (gen. slide no. HNUSEL-5557).

**Figure 5 insects-13-01107-f005:**
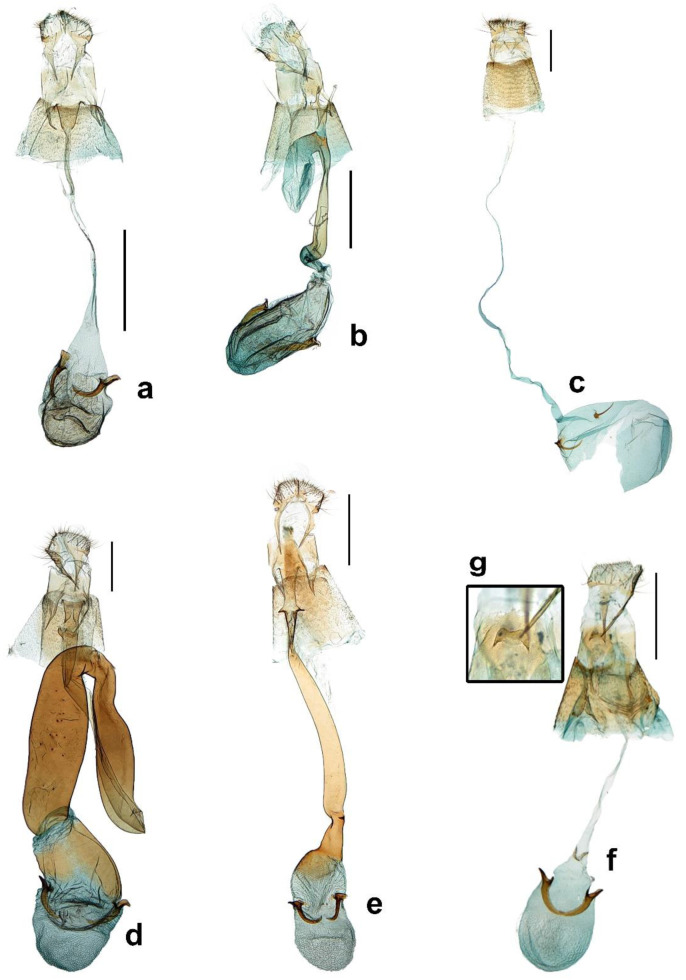
Female genitalia of *Caloptilia*. (**a**). *C. acericola* (gen. slide no. HNUSEL-5574); (**b**). *C. celtidis* (gen. slide no. HNUSEL-5613) (**c**). *C. kadsurae* (gen. slide no. HNUSEL-5558); (**d**). *C. monticola* (gen. slide no. HNUSEL-5587); (**e**). *C. recitata* (gen. slide no. HNUSEL-5543); (**f**). *C. soyella* (gen. slide no. HNUSEL-5597); (**g**). ditto, antrum (Scale bars: a–d = 5 mm).

## Data Availability

Data is contained within the article or Appendix A.

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
