# Peer review of "Taxonomic Review of the Genus Caloptilia Hübner, 1825 (Lepidoptera: Gracillariidae) with Descriptions of Three New Species and Seven Newly Recorded Species from Korea"

_insects, 2022, doi:10.3390/insects13121107_

Round 1

Reviewer 1 Report (New Reviewer)

The manuscript by Kim et al. provides important new information regarding the fauna of Caloptilia Hübner (Lepidoptera, Gracillariidae) from Korea, including data on geographic distribution and host plants, and the descriptions of three new species.

Despite all the information provided by the authors is extremely important to improve the understanding of the Gracillariidae from Korea, the manuscript needs major revisions before acceptation. I performed several annotations in the pdf file of the manuscript.

The terminology used for the female genitalia morphology is a bit confuse. I suggest the authors follow Landry (Landry B (2006) The Gracillariidae (Lepidoptera, Gracillarioidea) of the Galapagos Islands, Ecuador, with notes on some of their relatives. Revue Suisse de Zoologie 113, 437–485.) or any article about Caloptilia by Tosio Kumata to standardize all the terminology of their descriptions. They must indicate in Material and methods section the author they will follow in their descriptions.

Redescriptions should be included only if needed. They are not needed for each new country record. Redescriptions would be needed in cases in which original description are not good. If a given sex of a given species was previously unknown, thus the respective description must be included. Otherwise, descriptions or redescriptions are not needed. In contrast, all the figures of adults and genitalia provided by Kim et al. are extremely important for the paper. Perhaps a short diagnosis could be helpful.

The Discussion should be greatly improved. For instance, What about the new distribution records and host plant records provided by the authors? What about the host ranges of Caloptilia based on the information provided by the authors? What about future research of Caloptilia in Korea?

I am not a native English speaker, but the manuscript seems to require some serious language revision.

Author Response

Dear Reviewer 1,

Thank you for your comments. We read the comments carefully and revised our manuscript as follows.

Kind regards,

Bong-Kyu Byun,

On the behalf of the authors

Point 1: The terminology used for the female genitalia morphology is a bit confuse. I suggest the authors follow Landry (Landry B (2006) The Gracillariidae (Lepidoptera, Gracillarioidea) of the Galapagos Islands, Ecuador, with notes on some of their relatives. Revue Suisse de Zoologie 113, 437–485.) or any article about Caloptilia by Tosio Kumata to standardize all the terminology of their descriptions. They must indicate in Material and methods section the author they will follow in their descriptions.

Response 1: Thank you for your suggestion. We correct to the terminology of female genitalia according to Landry B (2006) as in the revised manuscript.

Point 2: Redescriptions should be included only if needed. They are not needed for each new country record. Redescriptions would be needed in cases in which original description are not good. If a given sex of a given species was previously unknown, thus the respective description must be included. Otherwise, descriptions or redescriptions are not needed. In contrast, all the figures of adults and genitalia provided by Kim et al. are extremely important for the paper. Perhaps a short diagnosis could be helpful.

Response 2: Thank you for your suggestion. We delete the duplicated redescription compared to the original description. But, in case of Caloptilia recitata (Meyrick, 1918) and Caloptilia soyella (van Deventer, 1904), there were only description for the adult of the former and for male genitalia of the latter were provided in the original description respectively. So, we inserted the description of the seasonal forms of the adults with the genitalic structures for the former and female genitalia for the latter.

Point 3: The Discussion should be greatly improved. For instance, What about the new distribution records and host plant records provided by the authors? What about the host ranges of Caloptilia based on the information provided by the authors? What about future research of Caloptilia in Korea?

Response 3: We revised the discussion in general as the reviewer’s comments as possible as we can.

Point 4: I am not a native English speaker, but the manuscript seems to require some serious language revision.

Response 4: To improve the quality of this manuscript, we conducted to the language revision by the native speaker as attached certificate.

Point 5:  Revision report for the reviewer’s correction directly marked in the PDF file

Reveiwer's suggestion

Authors response

1. Why "(A)"?

The (A) is the abbreviations for the Abdomen.

2. The segments VII and VIII of the male abdomen of Caloptilia are mainly membranous. Tergum and sternum of these segments can be at least slightly developed. Please see examples in Landry B. 2006. The Gracillariidae (Lepidoptera, Gracillarioidea) of the Galapagos Islands, Ecuador, with notes on some of their relatives. Revue Suisse de Zoologie 113, 437–485.

We checked the references that Landry (2006). And then, we correct to sclerotized sternite on A8.

3. Please separate the two words

Yes, we revised the manuscript.

4. I think you must delete "of"

Yes, we revised the manuscript.

5. You wish indicate "broadened near the corpus bursae"???

Yes, we revised 'near the' from 'neck of'.

6. Yes, endemic to Korea, but until now the geographic distribution of this micromoth is restricted to a specific area of Korea (based on the holotype data). Perhaps you could indicate some specific biogeographic area of Korea??

Yes, we added the detail region.

7. This is almost the same that Diagnosis. Do you have any other aspect to add here?

We wrote the “remarks part” by adding more detail morphological characteristic to diagnosis part. So, we think there is no point to add here aspect description.

8. Line 128, news

Yes, we deleted “s” .

9. Are you sure?? I think this is only membrane, not juxta

Thank you for your comment. But, I think it is a membranous juxta.

10. Perhaps you are calling antrum to sternum VIII??? I don´t understand this part of the description.

Thank you. We rechecked the part with microscope again, it is a part of the sternum VIII. So, we correct them.

11. Please more detail

Yes, we added the detail locality.

12. Do you know if other members of Caloptilia feed on this plant family and genus??

Yes, we found only this host plant for the species during the study.

13. As in the first species, Do you have some other aspect to include here??

Yes, we have no other aspect to include here.

14. I can't see the antrum in the Figure 4c. It would be possible to show detail (for this and for other figures)??

Sorry about that, I think you can find it in the revised manuscript file.

15. Please see comment in the first species.

Yes, we added the detail locality.

16. In general, these Remarks resemble much the Diagnosis. I think you could include other interesting aspects.

We wrote the remarks part more detail than diagnosis. In “diagnosis part”, it include the diagnostic character from the allied species.

17. This comparison is not, as the male genitalia of this new species is unknown.

Sorry about that. It was mistakenly included. We deleted it in the manuscript.

18. This appears to be an mountage effect.

Thank you. We rechecked the slide. It appears to be a characteristic of Caloptilia species, and also appears to be originally folded other than a mounting effect.

19. The coiled part appears to be membranous in the Figure 4e.

Yes, coiled part of ductus bursae is membranous. so, we revised the sentences. 

20. The male genitalia were described by Kumata (1966)

We correct "unknown" as "see Kumata 1966".

21. Please compare with description by Kumata (1966) to check agreement in morphological terms.

This re-description was deleted it

22. Kumata (1966) indicates that the ductus bursae is weakly sclerotized on caudal 1/3, strongly sclerotized and thickened on cephalic 2/3

We re-checked the paper by Kumata (1966) and the specimen we examined. Description of ductus bursae revised to “strongly sclerotized”.

23. I see two signa

Thank you. We corrected it as “singa”.

24. Kumata (1966) indicated that the corpus bursae is membranous with two large, sclerotized blotched on caudal half

Thank you. We deleted the description, which was included in the original description.

25. Yes, information regarding host plants of Gracillariidae is very important. However, it is not a rule that all the species are strictly host specific. Polyphagy is known for some species of Gracillariidae.

Yes, you are correct. So, we modified the part from the manuscript.

Reviewer 2 Report (New Reviewer)

This paper describes three new species and seven newly recorded species for Korea.The paper is very well illustrated and contains helpful information for identification of Korean Caloptilia. The paper represents an important contribution to the knowledge of Korean Gracilalriidae and the data seems to support their conclusions. Please see below some comments for authors to consider:

1.     Please add a table summarizing the host plant use of all Korean Caloptilia species. This could be included as an electronic appendix.

2.     Please add an annotated checklist of all Korean Caloptilia species. This could also be included as an electronic appendix.

3.     The manuscript needs extensive revision for language and grammar.

4.     It is a pity authors did not DNA barcoded at least the new species. It is an standard procedure. Could authors explain why?

Author Response

Dear Reviewer 2,

Thank you for your comments. We read the comments carefully and revised our manuscript as follows.

Kind regards,

Bong-Kyu Byun,

On the behalf of the authors

Point 1: Please add a table summarizing the host plant use of all Korean Caloptilia species. This could be included as an electronic appendix.

 Response 1: We worte it as attached Appenidx 1 for Korean Caloptilia species under the title “Host plants of the Korean Caloptilia species”.

Point 2: Please add an annotated checklist of all Korean Caloptilia species. This could also be included as an electronic appendix.

 Response 2: We worte it as attached Appenidx 2 under the title “Annotataed checklist of Korean Caloptilia species”.

Point 3:The manuscript needs extensive revision for language and grammar.

Response 3: To improve the quality of this manuscript, we conducted to the language revision by the native speaker as attached certificate.

Point 4: It is a pity authors did not DNA barcoded at least the new species. It is an standard procedure. Could authors explain why?

Response 4: Yes, it is true. In fact, we tried to extact the COI gene during the study, but it was failed due to the unknown reason. But, the new species are valid. During the course, we visited the major collection of the Gracillariidae, e.g. Microlepidoptera collection, the Natural History Museum, London, and Entomological collection, Hokkaido University, Sapporo, for the comparison of the type specimens with the new species and allied species as possible as we can.

Round 2

Reviewer 1 Report (New Reviewer)

Dear Authors,

Thank you very much for responding my comments and suggestions.

I think the manuscript is almost ready to be accepted. I included only three very minor observations in the revised version. For this reason I marked Minor Revisions.

Congratulations for this contribution to the improvement to the Celoptilia systematics.

Best regards,

Héctor

This manuscript is a resubmission of an earlier submission. The following is a list of the peer review reports and author responses from that submission.

Round 1

Reviewer 1 Report

It is difficult to discover new Lepidoptera species. Kim and his colleagues' description of three new species and seven new records from Korea is indeed impressive, however I believe the must be improved and expanded. The figures in pdf are in very low resolution, I can not find anything usefull. the most important thing is authors did not compare their new erection to their relate species, at leaset they should compare the new species with the type species and other species lived in the same faunae, besides, the diagnosis should be concise, the synamoporhies are welcome, or a unique combination characters, these are basic parts of a taxonomic work. Further, wing venation is very important of classification in Lepidoptera, but authors didn't provide any description on wing venation for new erection. In general,  this manuscript doesnt reach the average level of a taxonomic paper.

Author Response

Response to Reviewer 1 Comments

Reviewr 1

It is difficult to discover new Lepidoptera species. Kim and his colleagues' description of three new species and seven new records from Korea is indeed impressive, however I believe the must be improved and expanded. The figures in pdf are in very low resolution, I can not find anything usefull. the most important thing is authors did not compare their new erection to their relate species, at leaset they should compare the new species with the type species and other species lived in the same faunae, besides, the diagnosis should be concise, the synamoporhies are welcome, or a unique combination characters, these are basic parts of a taxonomic work. Further, wing venation is very important of classification in Lepidoptera, but authors didn't provide any description on wing venation for new erection. In general,  this manuscript doesnt reach the average level of a taxonomic paper.

Respond 1: Thank you for your helpful advice for our manuscript. We carefully revised and uploaded our manuscript as follwing corrections.

  1. In case of the figures, we uploaded the new image files as high resolution.
  2. In the “Diagnosis” part, we revised them as concise version.
  3. Also, we added the explantion of wing venation for three new species in the description part.

Reviewer 2 Report

Dear authors

This article is a very nice taxonomic review on the genus Caloptilia. It contains new species and new recorded species from Korea, and the novelty of this article is very high. This article must contribute to the accumulation of information of Korean moth fauna.

However, to write an article better, some corrections are needed as follows. This manuscript needs minor revision.

<Major points>

1)

You use ‘wing span’ as the size of the moth in the manuscript. But, the number of ‘wing span’ is not a fixed value, the word is not unsuitable for explanation of size.

You had better use ‘forewing length’ which is a fixed value. Please, measure the forewing length of each species.

2)

Would you like to propose Korean names to new species?

<Minor points>

L42: genus => the genus

L51-53: C. => C. (italic)

L54: this new species => these new species

L225: Kumata => Kumata (not italic)

L273: dentata => dentata (italic)

L570 and 571 => please exchange in chronological order.

L570 and 571: Please insert ‘:’ between specific name and author.

L572: Please insert ‘Caloptilia…’ in the synonymic list. The current genus name has been altered from the original description.

L612: Please insert ‘Caloptilia…’ in the synonymic list. The current genus name has been altered from the original description.

L713 and 714: Please insert the name of the original description in the synonymic list.

L714 and 715: Please insert ‘:’ between specific name and author.

L851: 2017 => 2015

L944: Please insert the current specific name in the synonymic list.

L973: Please insert the current specific name in the synonymic list.

Author Response

Response to Reviewer 2 Comments

 <Major points>

 Point 1: You use ‘wing span’ as the size of the moth in the manuscript. But, the number of ‘wing span’ is not a fixed value, the word is not unsuitable for explanation of size.

You had better use ‘forewing length’ which is a fixed value. Please, measure the forewing length of each species. -->We corrected all of ‘wing span’ as ‘forewing length’ with new measurements in the manuscript.

Point 2: Would you like to propose Korean names to new species? --> If the editor accept, we would like to propose the Korean names to new speices as the revised manuscript.

<Minor points> --> All of the “minor points” raised by the reviewer is corrected as the revised manuscript.

Point 3: L42: genus => the genus

Point 4: L51-53: C. => C. (italic)

Point 5: L54: this new species => these new species

Point 6: L225: Kumata => Kumata (not italic)

Point 7: L273: dentata => dentata (italic)

Point 8: L570 and 571 => please exchange in chronological order.

Point 9: L570 and 571: Please insert ‘:’ between specific name and author.

Point 10: L572: Please insert ‘Caloptilia…’ in the synonymic list. The current genus name has been altered from the original description.

Point 11: L612: Please insert ‘Caloptilia…’ in the synonymic list. The current genus name has been altered from the original description.

Point 12: L713 and 714: Please insert the name of the original description in the synonymic list.

Point 13: L714 and 715: Please insert ‘:’ between specific name and author.

Point 14: L851: 2017 => 2015

Point 15: L944: Please insert the current specific name in the synonymic list.

Point 16: L973: Please insert the current specific name in the synonymic list.

Round 2

Reviewer 1 Report

Dear authors:

I would strongly suggest to make a detailed comparison among the new species and its relatives in "remarks" part, you have to exhibit why you assign the species to this genus. This is very necessary to make your classification work justified and make it easier for readers to understand the difference between the new species and the known species.

Author Response

Response to Reviewer 1 Comments

Reviewer 1

Dear authors:

I would strongly suggest to make a detailed comparison among the new species and its relatives in "remarks" part, you have to exhibit why you assign the species to this genus. This is very necessary to make your classification work justified and make it easier for readers to understand the difference between the new species and the known species.

Respond to the reviewer comments:

Thank you for your helpful comments. We check the specimens carefully and rewrite the detalied comparison between the new species and its allied species in the “Remarks” part as in the revised mauscript. We hope it will be acceptable by the reviewers.

Reviewer 2 Report

Dear authors

Thank you for correction.

Your corrections are appropriate.

I feel that the manuscript can be acceptable.

Author Response

Response to Reviewer 2 Comments

Reviewer 2

Dear authors

Thank you for correction. Your corrections are appropriate.

I feel that the manuscript can be acceptable.

-> Thank you very much for your positive consideration on our paper as acceptable.
